# Are cause of death data fit for purpose? evidence from 20 countries at different levels of socio-economic development

Kim Moesgaard Iburg[1]*, Lene Mikkelsen[2], Tim Adair[2], Alan D. Lopez[2]

**1** Institute of Public Health, Aarhus University, Aarhus, Denmark, **2** Global Burden of Disease Group, University of Melbourne, Melbourne, Australia

* kmi@ph.au.dk

## Abstract

### Background and objective

Many countries have used the new ANACONDA (Analysis of Causes of National Death for Action) tool to assess the quality of their cause of death data (COD), but no cross-country analysis has been done to verify how different or similar patterns of diagnostic errors and data quality are in countries or how they are related to the local cultural or epidemiological environment or to levels of development. Our objective is to measure whether the usability of COD data and the patterns of unusable codes are related to a country's level of socio-economic development.

### Methods

We have assessed the quality of 20 national COD datasets from the WHO Mortality Database by assessing their completeness of COD reporting and the extent, pattern and severity of garbage codes, i.e. codes that provide little or no information about the true underlying COD. Garbage codes were classified into four groups based on the severity of the error in the code. The Vital Statistics Performance Index for Quality (VSPI(Q)) was used to measure the overall quality of each country's mortality surveillance system.

### Findings

The proportion of 'garbage codes' varied from 7 to 66% across the 20 countries. Countries with a high SDI generally had a lower proportion of high impact (i.e. more severe) garbage codes than countries with low SDI. While the magnitude and pattern of garbage codes differed among countries, the specific codes commonly used did not.

### Conclusions

There is an inverse relationship between a country's socio-demographic development and the overall quality of its cause of death data, but with important exceptions. In particular, some low SDI countries have vital statistics systems that are as reliable as more developed countries. However, in low-income countries, where most people die at home, the proportion

**Data Availability Statement:** The data underlying the results presented in the study are available in the WHO Mortality Database (apps.who.int/healthinfo/statistics/mortality/whodpms/) with the

following selected variables: Country, Reference Year, ICD-10 code, 5-years Age Group, Sex, and Number of deaths. Data were also obtained from the World Population Prospect 2017, UN Population Division (population.un.org/wpp/Download/Standard/Population/) with the following selected variables: Country, Reference Year, 5-years Age Group, Sex, and Population number.

**Funding:** This study was funded under an award from Bloomberg Philanthropies to the University of Melbourne to support the Data for Health Initiative: https://www.bloomberg.org/program/public-health/data-health/ Grant number not applicable. The funders had no role in study design, data collection and analysis, decision to publish, or preparation of the manuscript.

**Competing interests:** The authors have declared that no competing interests exist.

of unusable codes often exceeds 50%, implying that half of all cause-specific mortality data collected is of little or no use in guiding public policy. Moreover, the cause of death pattern identified from the data is likely to seriously under-represent the true extent of the leading causes of death in the population, with very significant consequences for health priority setting. Garbage codes are prevalent at all ages, contrary to expectations. Further research into effective strategies deployed in these countries to improve data quality can inform efforts elsewhere to improve COD reporting systems.

## Introduction

A key source of evidence for targeting health interventions to improve population health is high quality cause of death (COD) data that reliably document trends in mortality for different diseases and injuries [1]. However, several studies have demonstrated that policy and practice in many countries are based on data that are far from accurate [2–6]. In order to target efforts to improve the utility of COD data for policy, it is important to first understand what the key diagnostic errors are.

A major problem with COD data is poor cause of death certification practices that result in 'garbage codes', i.e. codes that provide little or no information about the true underlying cause of death [7]. Garbage codes include what are often called 'Ill-defined" causes, but encompass a larger universe of uninformative diagnoses. The major consequence of garbage codes is that they obscure the true mortality pattern in a population. For example, if a death certificate only states septicemia as the cause of death, there is no way of knowing whether this resulted, for example, from an infected wound following an accident, from a post-operative amputation due to diabetes, or from meningitis or pneumonia, each of which would require different preventive strategies. If the underlying cause that led to septicemia is not indicated on the death certificate, public policy to prevent these deaths would be misinformed, potentially leading to inefficient resource allocation to prevent them.

COD data provide the essential health intelligence for health policies across countries at various levels of socio-economic development. Our premise is that a better understanding of garbage codes, i.e. their levels and patterns in countries at different stages of socio-economic development, will help to target improvements in COD reporting systems. In this study, we investigate whether the usability of COD data and the patterns of garbage codes are related to a country's socio-economic development using the ANACONDA software tool [8, 9, S1 File] to assess the quality of 20 national COD datasets. Several countries have used this tool to assess how fit for purpose their data are [10–12], but there has not been any cross-country analysis of data quality across a range of socio-economic development levels and COD reporting systems using the common ANACONDA framework.

The implication of our findings for public policy to improve population health is that if the relationship is found to be very weak, or non-existent, then efforts to improve national Civil Registration and Vital Statistics systems can expect to make significant progress towards improving the evidence base for policy without depending on further socio-economic development.

## Data and methods

We carried out a cross-sectional study using publicly available data from the WHO Mortality Database [13], which contains COD data reported by its Member States. The 20 countries

were selected on the basis that they used ICD-10 [14], had provided data to WHO for a relatively recent year (2012–16), were located in all major regions of the world and differed in levels of socio-economic development. Population data were taken from the UN World Population Prospects 2017 [15], with the youngest age group of 0–4 years divided into 0- and 1-4-years age groups using Sprague's interpolation [16].

We classified a country's level of development, using the Socio Demographic Index (SDI) score from the Global Burden of Disease Study (GBD), into three levels: High, Middle and Low. The SDI is a summary measure of development expressed on a scale from 0 to 1 taking into account the total fertility rate, years of schooling, and gross national income [17]. For the lowest SDI level, the WHO Mortality Database only contained the few countries we selected; for the Middle and High SDI levels we selected countries with recent data from different regions of the world. Our study included 4 Low SDI countries, 10 Middle SDI, and 6 High SDI countries.

On the basis of the country specific ICD-10 codes used in GBD 2017 [18], the most severe certification and coding errors that can mislead policy and public health planning were identified and categorized into four groups. The four-tier garbage code typology used in ANACONDA is based on the premise that some garbage codes are worse than others depending on how serious their impact is for guiding or misguiding policy debates and will thus likely impact disease and injury control strategies differently [19]:

- **Level 1 (very high)–codes with serious policy implications.** These are causes for which the true underlying COD could in fact belong to any of three broad cause group (i.e. it is impossible to establish whether the true cause was a communicable disease, a non-communicable disease or because of an injury, a good example being 'septicaemia' reported as the underlying cause of death). These are the most serious mis-diagnoses among the universe of unusable codes, since they could potentially grossly misinform understanding of the extent of epidemiological transition in the population.

- **Level 2 (high)–codes with substantial implications for policy.** These are causes for which the true underlying COD is likely to belong to one or two of the three broad cause groups; for example, 'essential (primary) hypertension'.

- **Level 3 (medium)–codes with important implications for policy.** These are causes for which the true underlying COD is likely to be within the same ICD chapter, for example, 'unspecified cancer', and thus are of some policy value.

- **Level 4 (low)–codes with limited implications for policy.** These are diagnoses for which the true underlying COD is likely to be confined to a single disease or injury category (e.g. unspecified stroke would still be assigned as a stroke death, and not to some other disease category). The implications of unusable causes classified at this level will therefore be much less important for public policy, but a more specific code would have increased their utility for specific public health analyses.

A full list of the composition of specific ICD-10 garbage codes for each of the four severity levels is given in S2 File.

Given the considerable differences in population age structure between countries at high and low levels of socio-economic development, we age standardised the pattern of garbage codes. The point of age-standardising was to investigate whether countries with a comparatively old age structure, and hence relatively high average age at death, might expect to have a greater fraction of garbage codes simply because of the higher likelihood of multiple co-morbidity in the elderly. We used the global age distribution of deaths from the latest GBD Study as the standard [20].

In addition to diagnostic accuracy, the ability of any dataset to describe the true mortality pattern in a population also depends on how complete it is, both in terms of capturing all deaths that occur, and in assigning each a COD. Completeness of the COD reporting (i.e. the percentage of actual deaths in a population that are assigned a COD) for each of the 20 countries was calculated using the Adair-Lopez empirical method incorporated into ANACONDA [21]. The empirical method models the relationship between the Crude Death Rate (CDR) and its principal determinants, namely the age structure of the population and the overall level of mortality, as reflected by the level of child mortality. The predicted CDR based on these input variables for a population is then compared with the observed CDR to estimate death registration completeness. Given that the model was built largely from historical data where the levels of adult mortality and child mortality are closely correlated, the predictions of completeness for populations where this assumption is not valid, such as those severely affected by HIV, should be interpreted cautiously.

Datasets that are both incomplete and have a high proportion of garbage codes provide limited insight into the true health status of a population. We combined the proportion of unrecorded deaths with the amount of garbage codes to provide a summary measure of the utility of the data for policy. This indicator is particularly important when investigating data quality in countries with low completeness where the data available may only come from hospitals and other health facilities where diagnostic facilities and physician availability is greater, potentially over-stating the policy utility of the data.

A key output of any mortality surveillance system is a table showing the leading causes of death for the population. In countries where garbage codes are commonly assigned, they frequently appear among the 10 or 20 leading causes and can seriously impact the overall utility of the COD data. This is particularly the case when they permeate the top 10 leading causes and are "high impact", providing little or no useful information for policy.

The ANACONDA software tool specifically developed for assessing quality of mortality and COD data, was used to investigate each dataset (S1 File) to identify the pattern and extent of garbage codes in the data, their frequency among the leading causes, the completeness of the dataset and to provide an overall summary index of the quality of the output of the mortality data system, namely the Vital Statistics Performance Index for Quality (VSPI(Q)) [22].

## Results

The proportion of garbage codes in the 20 country datasets varied substantially by country, ranging from 7% to 66% (see Fig 1). While the relationship between SDI and amount of garbage codes in the data is broadly apparent, the relatively low $R^2$ (0.17) arises from the presence of outliers, particularly Uzbekistan, Kyrgyzstan, Nicaragua and Colombia. It is quite possible that specific certification and coding procedures have been introduced in these countries to avoid the use of garbage codes. If these countries are omitted, the strength of the inverse relationship is much more apparent. Further insights into the general characteristics of population and mortality of the selected countries can be found in S2 File.

The mean values for each SDI group indicate that, on average, countries of High SDI status had a lower (23.8%) proportion of garbage codes in their data than Middle (39.7%) and Low (40.0%) SDI countries (Table 1). Interestingly, there was large inter-country variation in the use of garbage codes within each SDI level. For example, of the High SDI countries, Finland had the lowest amount of garbage codes (7%) in their data, whereas in Japan (36%) and France (34%), the level was five times higher, affecting about one in three deaths. For the Middle SDI group, Argentina, Thailand and Tunisia had more than half of all CODs coded to a garbage code (in Thailand, close to 80% of these were high impact errors), while for other countries in

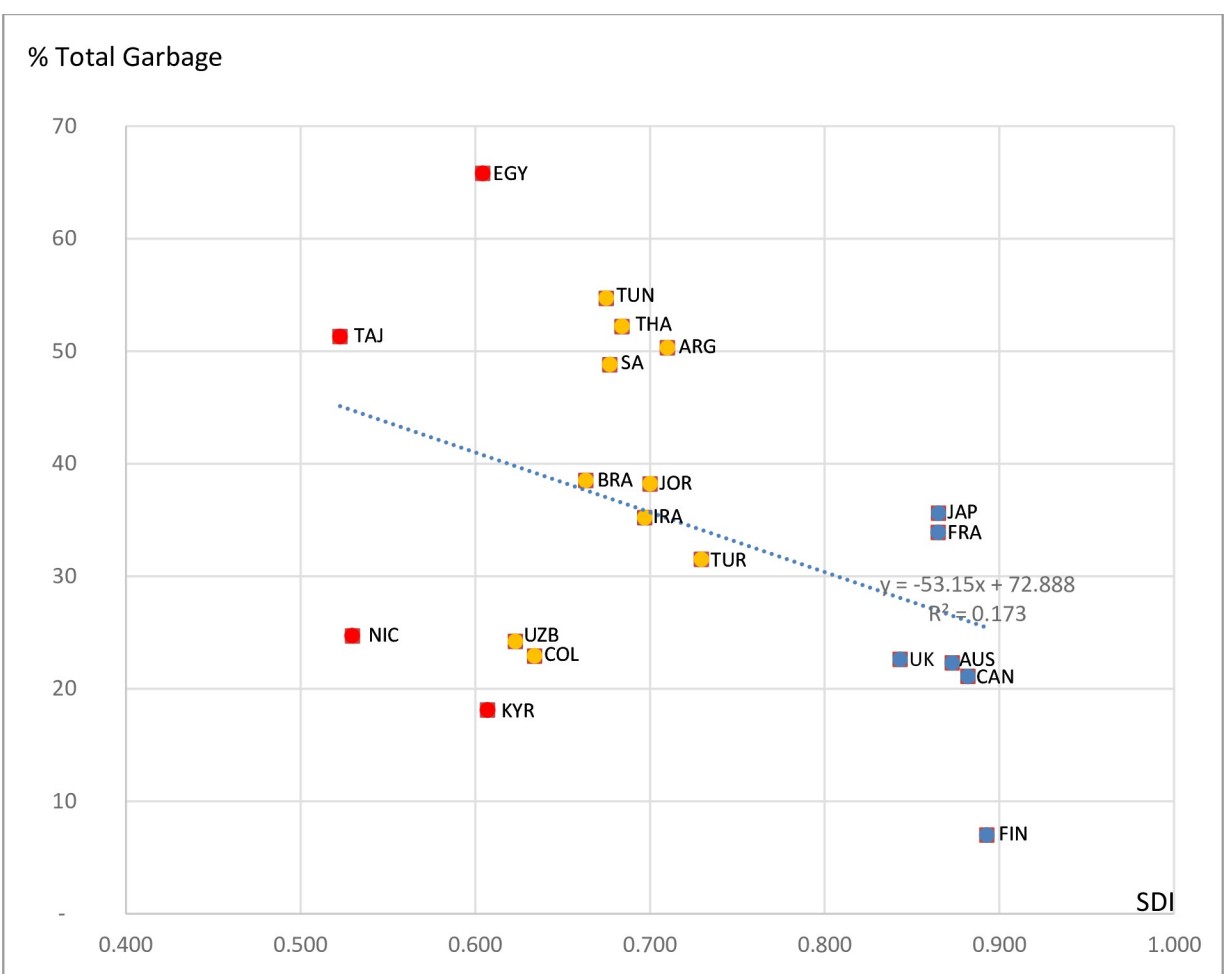

**Fig 1. Percentage of total garbage codes versus socio demographic index, selected countries.** High SDI, Middle SDI, Low SDI.

this group, notably Uzbekistan, Turkey and Colombia, the use of garbage codes was much less prominent. Surprisingly, among the Low SDI countries, Kyrgyzstan and Nicaragua had comparatively low levels of garbage, (18% and 25%, respectively), whereas in Egypt, at a similar SDI level, two-thirds of all deaths are coded to garbage codes and of these, over 80% were of high impact (Table 1).

Importantly, the fraction of high impact garbage codes ranged from a low of 6% (Finland) to 57% (Egypt), and for low impact codes, from 1% (Finland) to 18% (Brazil). Countries with high socio-demographic development had a lower proportion of high impact garbage codes (mean 16%) compared with Middle (26%) and Low (29%) SDI countries (Table 1). Ranking countries according to their percent of high impact garbage codes reveals a very substantial gap between the best and worst performing COD information systems and a surprising mixture of SDI levels (Fig 2). Kyrgyzstan has the second-best performing system, after Finland, with COD data in Colombia of almost equal quality to the UK, and that in Nicaragua falling between Australia and Canada. Uzbekistan, Turkey, Brazil and Jordan all assign less causes to high impact garbage codes than both Japan and France.

The impact of population age structure on the overall level of garbage codes varied across countries but was generally small, contrary to what might have been expected (Table 1). In Japan and Argentina, the age-adjusted fraction of garbage codes was 4–6% points lower than

**Table 1. Levels of garbage codes standardised by age and registration completeness for select countries ranked by their Socio-Demographic Index (SDI).**

| Country | Year | SDI value | High impact GC (%) | Low impact GC (%) | (a) Total GC (age-adj GC) (%) | (b) Completeness of COD Registration (%) | (c) Deaths of no policy value (%) |
|---|---|---|---|---|---|---|---|
| Finland | 2015 | 0.893 | 5.7 | 1.3 | 7.0 (8.0) | 99.2 | 7.7 |
| Canada | 2013 | 0.882 | 12.7 | 8.4 | 21.1 (22.1) | 100 | 21.1 |
| Australia | 2015 | 0.873 | 14.1 | 8.2 | 22.3 (22.0) | 100 | 22.3 |
| Japan | 2015 | 0.865 | 24.9 | 10.7 | 35.6 (29.3) | 98.4 | 36.6 |
| France | 2014 | 0.865 | 27.6 | 6.3 | 33.9 (38.7) | 99.2 | 34.4 |
| United Kingdom | 2015 | 0.843 | 11.2 | 11.4 | 22.6 (22.1) | 99.2 | 23.2 |
| **High SDI (mean)** | | **0.870** | **16.0** | **7.7** | **23.8 (23.7)** | **99.3** | **24.2** |
| Turkey | 2015 | 0.729 | 18.3 | 13.2 | 31.5 (30.2) | 95.2 | 34.8 |
| Argentina | 2015 | 0.710 | 34.1 | 16.2 | 50.3 (46.7) | 100 | 50.3 |
| Iran | 2015 | 0.700 | 26.6 | 8.6 | 35.2 (34.7 | 92.4 | 40.1 |
| Jordan | 2012 | 0.697 | 22.2 | 16.0 | 38.2 (38.7) | 43.8 | 72.9 |
| Thailand | 2016 | 0.684 | 41.2 | 11.0 | 52.2 (50.3) | 93.7 | 55.2 |
| South Africa | 2014 | 0.677 | 32.1 | 16.7 | 48.8 (56.0) | 94.9 | 51.4 |
| Tunisia | 2013 | 0.675 | 39.3 | 15.4 | 54.7 (53.9) | 68.7 | 68.9 |
| Brazil | 2015 | 0.663 | 21.0 | 17.5 | 38.5 (38.0) | 96.3 | 40.8 |
| Colombia | 2015 | 0.634 | 11.6 | 11.3 | 22.9 (22.7) | 97.4 | 24.9 |
| Uzbekistan | 2014 | 0.623 | 16.2 | 8.0 | 24.2 (24.8) | 85.4 | 35.3 |
| **Middle SDI (mean)** | | **0.679** | **26.3** | **13.4** | **39.7 (39.6)** | **86.8** | **47.5** |
| Kyrgyzstan | 2015 | 0.607 | 8.5 | 9.6 | 18.1 (17.7) | 93.2 | 23.7 |
| Egypt | 2015 | 0.604 | 56.7 | 9.1 | 65.8 (66.8) | 91.4 | 68.7 |
| Nicaragua | 2015 | 0.530 | 13.3 | 11.4 | 24.7 (26.0) | 91.9 | 30.8 |
| Tajikistan | 2016 | 0.523 | 37.8 | 13.5 | 51.3 (52.3) | 58.3 | 71.6 |
| **Low SDI (mean)** | | **0.566** | **29.1** | **10.9** | **40.0 (40.7)** | **83.7** | **48.7** |
| **Total Mean** | | **0.714** | **23.8** | **11.2** | **34.9** | **89.9** | **40.7** |

**SDI levels**

SDI three levels collapsed from GBD 2017 five SDI levels:

High = High

Middle = High middle + Middle

Low = Low + Low middle

**Garbage code (GC) levels**

High impact GC = Levels 1–3

Low impact GC = Level 4

Total GC = high + low

**Death of no value for policy**

(c) = (1-b) + (a*b)

This equation calculates unavailable deaths for policy; i.e. unregistered deaths plus deaths with a garbage code

the un-adjusted fraction, while in France and South Africa it increased by a similar amount (column **a**). Age-standardisation, therefore, had no impact on the mean level of garbage codes in each development category.

Age-standardisation, however, masks the age pattern of garbage coding, particularly its relative importance at younger adult ages where accurate and specific diagnoses are critical for guiding policies designed to prevent premature deaths. The perception that garbage codes are largely confined to deaths among the elderly due to the presence of co-morbidities at or around the time of death is not confirmed by the age-specific fractions of garbage codes shown

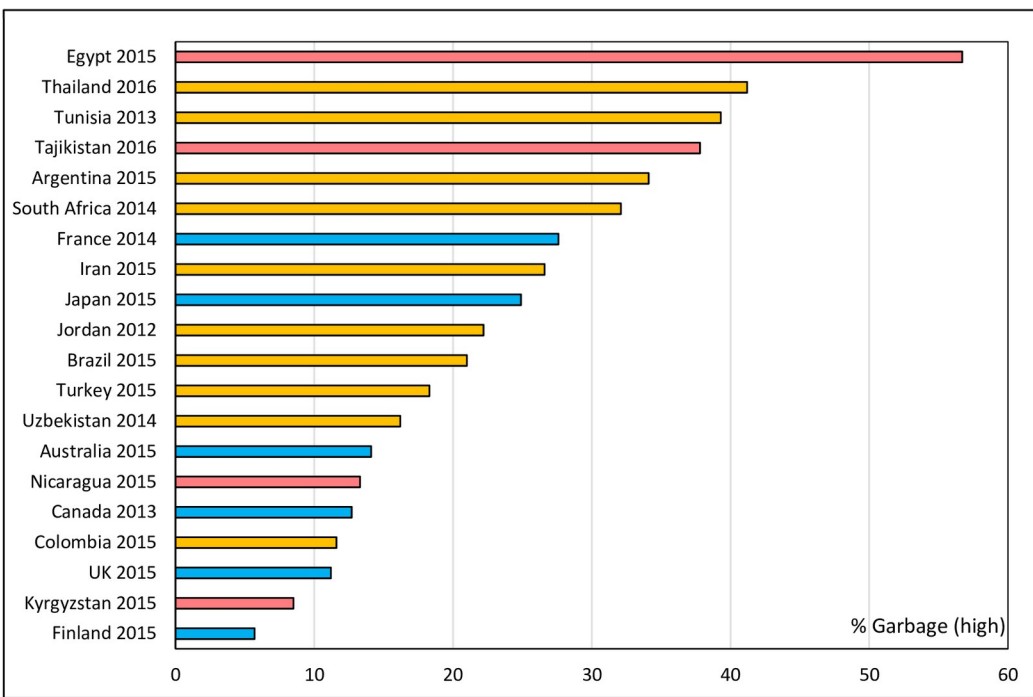

**Fig 2. Percentage of high impact garbage codes in total causes of death, selected countries, c. 2015.** High SDI, Middle SDI, Low SDI.

in Fig 3, with exact fractions reported in S3 File. Garbage codes are prevalent at all ages and often in similar proportions to what is observed for the 70+ age group. Indeed, in some highly developed countries (e.g. Finland, Canada, UK) the proportion of garbage codes is significantly higher at ages 20–49 for both sexes, than at older ages. Indeed, in Tunisia for males, and

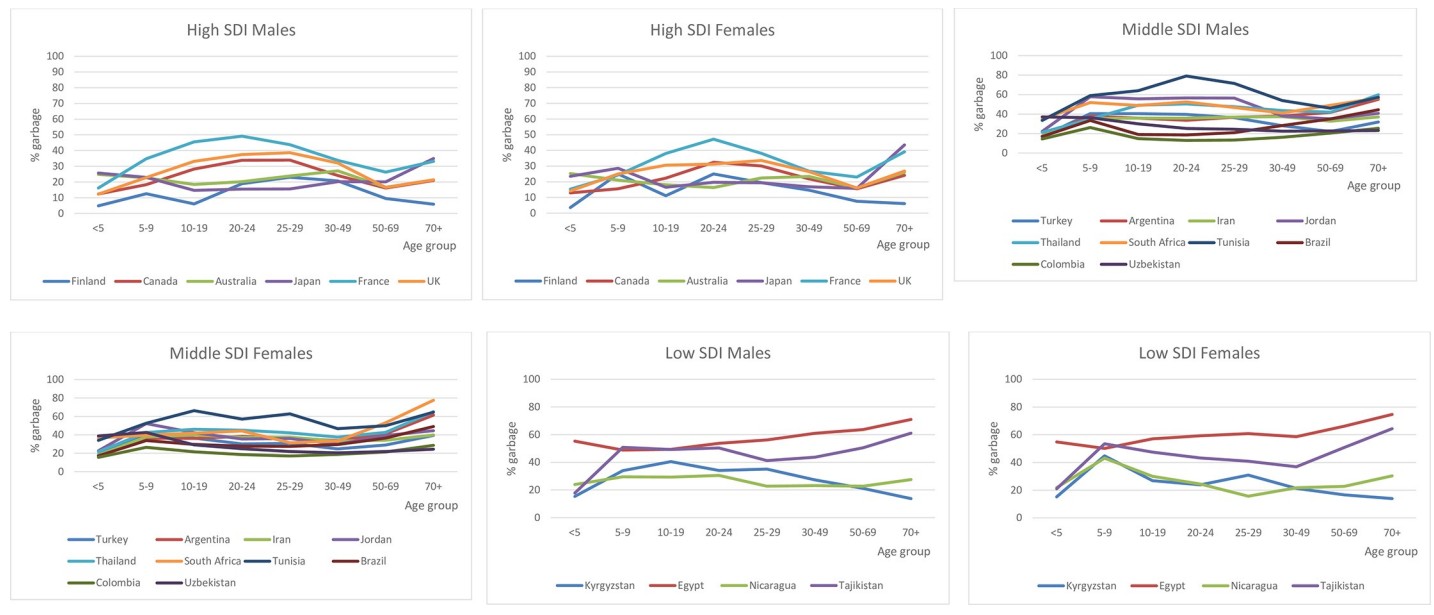

**Fig 3. Garbage codes as a percentage of all deaths, by age, selected countries.**

in Jordan and Kyrgyzstan for both sexes, this pattern is already evident from age 5; in Uzbekistan, garbage coding is more common for deaths of children and adolescents than at ages 70 and above.

Given that the utility of a country's mortality information system is reduced not only by the amount of garbage codes but also by how complete it is in terms of capturing all deaths, we measured their combined impact. For two countries, namely Jordan, and Tajikistan, the consolidated indicator of registration completeness and fraction of garbage codes was close to 75%, suggesting that useful information on only about one quarter of all deaths that occur in Jordan and Tajikistan is available for policy purposes (Col. C Table 1). Overall, the combined indicator showed the proportion of deaths for which information was either missing or of little or no value for guiding health policy was much higher in Low (49%) and Middle (48%) SDI countries compared to High (24%) SDI countries.

Table 2 shows the strong influence of garbage codes on the leading cause distribution when these are ranked, with high impact garbage codes marked in red and low impact in orange. The presence of High impact garbage codes among the 10 leading causes of death will substantially distort the true picture of what are the common COD that most people die from. Among High SDI countries only Japan and France have high impact garbage codes (for males) among the ten leading causes of death. Canada, Australia and UK had only low impact causes in this category, and Finland had neither. For the Middle and Low SDI countries there were many more high impact garbage codes listed among the leading COD, with Egypt having seven, Tajikistan five, and Thailand and Tunisia each with four. Colombia and Nicaragua had none. For females, Egypt had seven, Iran, Tunisia and Tajikistan five, with the remaining countries having between one and three.

Notwithstanding some variation in patterns and volume of garbage coding across countries, the most common garbage codes were remarkably similar. In particular, Other ill-defined and unspecified causes of death, Senility, Heart failure, Unspecified neoplasm, Septicemia, Respiratory failure, Unknown cause of death, Hypertension and Unspecified diabetes were observed across all SDI levels. In addition, Low SDI countries tended to report Atherosclerosis, Hepatic failure, Intracerebral hemorrhage and Unattended deaths, all garbage codes, as leading causes.

Fig 4 ranks countries according to a single consolidated summary measure of system performance, namely the Vital Statistics Performance Index for Quality (VSPI(Q)). Eleven countries scored 70 and above, a level where they could be considered as having well-functioning systems. Six of the remaining countries achieved scores that would classify them as having medium performing systems, with lower scores mostly arising from the high proportion of garbage codes that bias their COD distributions. Of the 20 countries, only Jordan, Tajikistan, and Tunisia were classified as having poorly functioning systems.

## Discussion

In general, one would expect that in High SDI countries, where all deaths are medically certified with good quality clinical and diagnostic services, comparatively few deaths would be assigned unusable (garbage) codes, and certainly much less than in countries where such services are less common. Our findings based on an analysis of 20 COD datasets from across the world produced evidence both for and against this hypothesis. France and Japan do not seem to have more reliable COD data to guide policy than Turkey, Colombia, Kyrgyzstan and Nicaragua. In France, 1 in 3 deaths is assigned a garbage code, with 9% of all deaths being certified as due to "Other ill-defined and unspecified causes of death" (R99), Heart failure (I50.9) or Respiratory arrest (R09.2). Similarly, in Japan, "old age" (Senility R54) is a commonly assigned cause of death accounting for one fifth of all garbage codes. Other research has found that

Table 2. Top-10 leading causes of death by country and sex (high impact garbage codes marked in red; low impact ones in yellow).

| Males / Country | Rank 1 | 2 | 3 | 4 | 5 | 6 | 7 | 8 | 9 | 10 |
|---|---|---|---|---|---|---|---|---|---|---|
| Finland | Atherosclerotic heart disease | Alzheimer's disease with late onset | Malignant neoplasm of prostate | Acute myocardial infarction, unspecified | Alzheimer's disease, unspecified | Upper lobe, bronchus or lung | Alcoholic cirrhosis of liver | Chronic obstructive pulmonary disease with acute lower respiratory infection | Sequelae of cerebral infarction | Chronic ischaemic heart disease, unspecified |
| Canada | Bronchus or lung, unspecified | Acute myocardial infarction, unspecified | Atherosclerotic heart disease | Unspecified dementia | Malignant neoplasm of prostate | Stroke, not specified as haemorrhage or infarction | Chronic obstructive pulmonary disease, unspecified | Colon, unspecified | Pneumonia, unspecified | Pancreas, unspecified |
| Australien | Bronchus or lung, unspecified | Acute myocardial infarction, unspecified | Chronic ischaemic heart disease, unspecified | Malignant neoplasm of prostate | Unspecified dementia | Chronic obstructive pulmonary disease, unspecified | Atherosclerotic heart disease | Stroke, not specified as haemorrhage or infarction | Pancreas, unspecified | Intentional self-harm by hanging, strangulation and suffocation, unspecified place |
| Japan | Pneumonia, unspecified | Bronchus or lung, unspecified | Stomach, unspecified | Heart failure, unspecified | Senility | Acute myocardial infarction, unspecified | Pneumonitis due to food and vomit | Liver cell carcinoma | Sequelae of cerebral infarction | Cerebral infarction, unspecified |
| United Kingdom | Bronchus or lung, unspecified | Acute myocardial infarction, unspecified | Chronic ischaemic heart disease, unspecified | Unspecified dementia | Malignant neoplasm of prostate | Atherosclerotic heart disease | Chronic obstructive pulmonary disease with acute lower respiratory infection | Pneumonia, unspecified | Stroke, not specified as haemorrhage or infarction | Bronchopneumonia, unspecified |
| France | Bronchus or lung, unspecified | Other ill-defined and unspecified causes of mortality | Malignant neoplasm of prostate | Acute myocardial infarction, unspecified | Respiratory arrest | Chronic ischaemic heart disease, unspecified | Heart failure, unspecified | Colon, unspecified | Alzheimer's disease, unspecified | Pancreas, unspecified |
| Turkey | Acute myocardial infarction, unspecified | Bronchus or lung, unspecified | Heart failure, unspecified | Chronic obstructive pulmonary disease, unspecified | Atherosclerotic heart disease | Cerebrovascular disease, unspecified | Chronic obstructive pulmonary disease with acute lower respiratory infection | Pneumonia, unspecified | Stomach, unspecified | Alzheimer's disease with late onset |
| Argentina | Pneumonia, unspecified | Acute myocardial infarction, unspecified | Heart failure, unspecified | Bronchus or lung, unspecified | Other ill-defined and unspecified causes of mortality | Stroke, not specified as haemorrhage or infarction | Septicaemia, unspecified | Malignant neoplasm of prostate | Chronic obstructive pulmonary disease, unspecified | Colon, unspecified |
| Iran | Acute myocardial infarction | Cerebral infarction | Hypertensive heart disease with (congestive) heart failure | Cardiac arrest | Unspecified diabetes mellitus | Chronic ischaemic heart disease | Hypertensive heart disease | Malignant neoplasm of stomach | Complications and ill-defined descriptions of heart disease | Malignant neoplasm of bronchus and lung |
| Jordan | Acute myocardial infarction, unspecified | Unspecified transport accident | Other ill-defined and unspecified causes of mortality | Unspecified diabetes mellitus without complications | Cerebral infarction, unspecified | Bronchus or lung, unspecified | Essential (primary) hypertension | Atherosclerotic heart disease | Pneumonia, unspecified | Heart failure, unspecified |

(*Continued*)

Table 2. (Continued)

| Country | 1 | 2 | 3 | 4 | 5 | 6 | 7 | 8 | 9 | 10 |
|---|---|---|---|---|---|---|---|---|---|---|
| Thailand | Other ill-defined and unspecified causes of mortality | Pneumonia, organism unspecified | Senility | Malignant neoplasm of liver and intrahepatic bile ducts | Other septicaemia | Malignant neoplasm of bronchus and lung | Intracerebral haemorrhage | Acute myocardial infarction | Chronic ischaemic heart disease | Unspecified event, undetermined intent |
| South Africa | Other ill-defined and unspecified causes of mortality | Respiratory tuberculosis, not confirmed bacteriologically or histologically | Pneumonia, organism unspecified | Unspecified diabetes mellitus | Exposure to unspecified factor | Stroke, not specified as haemorrhage or infarction | Human immunodeficiency virus [HIV] disease resulting in infectious and parasitic diseases | Other viral diseases, not elsewhere classified | Diarrhoea and gastroenteritis of presumed infectious origin | Heart failure |
| Tunisia | Other ill-defined and unspecified causes of mortality | Bronchus or lung, unspecified | Acute myocardial infarction, unspecified | Stroke, not specified as haemorrhage or infarction | Unspecified event, undetermined intent, unspecified place | Unspecified diabetes mellitus with other specified complications | Respiratory arrest | Senility | Cerebral infarction due to unspecified occlusion or stenosis of cerebral arteries | Person injured in unspecified motor-vehicle accident, traffic |
| Brazil | Acute myocardial infarction, unspecified | Pneumonia, unspecified | Other ill-defined and unspecified causes of mortality | Stroke, not specified as haemorrhage or infarction | Assault by other and unspecified firearm discharge, street and highway | Bronchus or lung, unspecified | Malignant neoplasm of prostate | Unspecified diabetes mellitus without complications | Essential (primary) hypertension | Unattended death |
| Colombia | Acute myocardial infarction, unspecified | Assault by other and unspecified firearm discharge, street and highway | Chronic obstructive pulmonary disease, unspecified | Pneumonia, unspecified | Stomach, unspecified | Malignant neoplasm of prostate | Chronic obstructive pulmonary disease with acute lower respiratory infection | Bronchus or lung, unspecified | Assault by other and unspecified firearm discharge, unspecified place | Chronic renal failure, unspecified |
| Uzbekistan | Chronic ischaemic heart disease | Hypertensive heart disease | Angina pectoris | Acute myocardial infarction | Fibrosis and cirrhosis of liver | Intracerebral haemorrhage | Other acute ischaemic heart diseases | | | Atherosclerosis |
| Kyrgyzstan | Atherosclerotic heart disease | Stroke, not specified as haemorrhage or infarction | Other and unspecified cirrhosis of liver | Acute myocardial infarction, unspecified | Other ill-defined and unspecified causes of mortality | Other specified chronic obstructive pulmonary disease | Atherosclerotic cardiovascular disease, so described | Chronic ischaemic heart disease, unspecified | Stomach, unspecified | Intentional self-harm by hanging, strangulation and suffocation, home |
| Egypt | Heart failure | Essential (primary) hypertension | Cardiac arrest | Fibrosis and cirrhosis of liver | Hepatic failure, not elsewhere classified | Intracerebral haemorrhage | Respiratory failure, not elsewhere classified | Acute myocardial infarction | Senility | Atherosclerosis |
| Nicaragua | Acute myocardial infarction, unspecified | Chronic renal failure, unspecified | Pneumonia, unspecified | Stroke, not specified as haemorrhage or infarction | Chronic obstructive pulmonary disease, unspecified | Person injured in unspecified motor-vehicle accident, traffic | Other and unspecified cirrhosis of liver | Alcoholic cirrhosis of liver | Non-insulin-dependent diabetes mellitus with renal complications | Malignant neoplasm of prostate |
| Tajikistan | Essential (primary) hypertension | Stroke, not specified as haemorrhage or infarction | Senility | Other acute ischaemic heart diseases | Acute myocardial infarction | Other ill-defined and unspecified causes of mortality | Atherosclerosis | Chronic ischaemic heart disease | Unspecified diabetes mellitus | Complications and ill-defined descriptions of heart disease |
| **Females** | | | | | | | | | | |
| Country | Rank 1 | 2 | 3 | 4 | 5 | 6 | 7 | 8 | 9 | 10 |

(Continued)

**Table 2.** (Continued)

| Country | | | | | | | | | | |
|---|---|---|---|---|---|---|---|---|---|---|
| **Finland** | Alzheimer's disease with late onset | Atherosclerotic heart disease | Alzheimer's disease, unspecified | Hypertensive heart disease with (congestive) heart failure | Acute myocardial infarction, unspecified | Unspecified dementia | Chronic ischaemic heart disease, unspecified | Cerebral infarction, unspecified | Other Alzheimer's disease | Sequelae of cerebral infarction |
| **Canada** | Unspecified dementia | Bronchus or lung, unspecified | Acute myocardial infarction, unspecified | Atherosclerotic heart disease | Breast, unspecified | Stroke, not specified as haemorrhage or infarction | Alzheimer's disease, unspecified | Chronic obstructive pulmonary disease, unspecified | Pneumonia, unspecified | Colon, unspecified |
| **Australien** | Unspecified dementia | Acute myocardial infarction, unspecified | Bronchus or lung, unspecified | Chronic ischaemic heart disease, unspecified | Stroke, not specified as haemorrhage or infarction | Breast, unspecified | Alzheimer's disease, unspecified | Chronic obstructive pulmonary disease, unspecified | Atrial fibrillation and atrial flutter, unspecified | Pancreas, unspecified |
| **Japan** | Senility | Pneumonia, unspecified | Heart failure, unspecified | Bronchus or lung, unspecified | Pneumonitis due to food and vomit | Sequelae of cerebral infarction | Cerebral infarction, unspecified | Acute myocardial infarction, unspecified | Stomach, unspecified | Breast, unspecified |
| **United Kingdom** | Unspecified dementia | Bronchus or lung, unspecified | Chronic ischaemic heart disease, unspecified | Breast, unspecified | Stroke, not specified as haemorrhage or infarction | Acute myocardial infarction, unspecified | Alzheimer's disease, unspecified | Pneumonia, unspecified | Vascular dementia, unspecified | Chronic obstructive pulmonary disease with acute lower respiratory infection |
| **France** | Alzheimer's disease, unspecified | Breast, unspecified | Other ill-defined and unspecified causes of mortality | Unspecified dementia | Bronchus or lung, unspecified | Heart failure, unspecified | Respiratory arrest | Stroke, not specified as haemorrhage or infarction | Acute myocardial infarction, unspecified | Colon, unspecified |
| **Turkey** | Acute myocardial infarction, unspecified | Cerebral infarction | Cerebrovascular disease, unspecified | Alzheimer's disease with late onset | Hypertensive heart disease with (congestive) heart failure | Chronic ischaemic heart disease | Pneumonia, unspecified | Other ill-defined and unspecified causes of mortality | Sequelae of other and unspecified cerebrovascular diseases | Breast, unspecified |
| **Argentina** | Pneumonia, unspecified | Heart failure, unspecified | Unspecified diabetes mellitus without complications | Other ill-defined and unspecified causes of mortality | Breast, unspecified | Septicaemia, unspecified | Stroke, not specified as haemorrhage or infarction | Pneumonia, unspecified | Respiratory failure, unspecified | Colon, unspecified |
| **Iran** | Cerebral infarction, unspecified | Hypertensive heart disease with (congestive) heart failure | Cerebral infarction | Hypertensive heart disease | Cardiac arrest | Chronic ischaemic heart disease | Essential (primary) hypertension | Other ill-defined and unspecified causes of mortality | Senility | Complications and ill-defined descriptions of heart disease |
| **Jordan** | Hypertensive heart disease with (congestive) heart failure | Acute myocardial infarction, unspecified | Unspecified diabetes mellitus without complications | Acute myocardial infarction, unspecified | Essential (primary) hypertension | Breast, unspecified | Pneumonia, unspecified | Heart failure, unspecified | Atherosclerotic heart disease | Congenital malformation of heart, unspecified |
| **Thailand** | Other ill-defined and unspecified causes of mortality | Senility | Pneumonia, organism unspecified | Other septicaemia | Unspecified diabetes mellitus | Other degenerative diseases of nervous system, not elsewhere classified | Chronic renal failure | Malignant neoplasm of liver and intrahepatic bile ducts | Malignant neoplasm of bronchus and lung | Intracerebral haemorrhage |

(Continued)

**Table 2.** (Continued)

| | 1 | 2 | 3 | 4 | 5 | 6 | 7 | 8 | 9 | 10 | 11 |
|---|---|---|---|---|---|---|---|---|---|---|---|
| **South Africa** | Other ill-defined and unspecified causes of mortality | Unspecified diabetes mellitus | Respiratory tuberculosis, not confirmed bacteriologically or histologically | Stroke, not specified as haemorrhage or infarction | Pneumonia, organism unspecified | Diarrhoea and gastroenteritis of presumed infectious origin | Other viral diseases, not elsewhere classified | Heart failure | Human immunodeficiency virus [HIV] disease resulting in infectious and parasitic diseases | Essential (primary) hypertension | |
| **Tunisia** | Other ill-defined and unspecified causes of mortality | Stroke, not specified as haemorrhage or infarction | Senility | Acute myocardial infarction, unspecified | Unspecified diabetes mellitus with other specified complications | Unspecified event, undetermined intent, unspecified place | Breast, unspecified | Respiratory arrest | Cerebral infarction due to unspecified occlusion or stenosis of cerebral arteries | Heart failure, unspecified | |
| **Brazil** | Acute myocardial infarction, unspecified | Pneumonia, unspecified | Stroke, not specified as haemorrhage or infarction | Unspecified diabetes mellitus without complications | Breast, unspecified | Other ill-defined and unspecified causes of mortality | Essential (primary) hypertension | Alzheimer's disease, unspecified | Bronchus or lung, unspecified | Septicaemia, unspecified | |
| **Columbia** | Acute myocardial infarction, unspecified | Pneumonia, unspecified | Chronic obstructive pulmonary disease, unspecified | Breast, unspecified | Chronic obstructive pulmonary disease with acute lower respiratory infection | Bronchus or lung, unspecified | Stomach, unspecified | Hypertensive heart disease with (congestive) heart failure | Cervix uteri, unspecified | Essential (primary) hypertension | |
| **Uzbekistan** | Chronic ischaemic heart disease | Hypertensive heart disease | Angina pectoris | Fibrosis and cirrhosis of liver | Intracerebral haemorrhage | Acute myocardial infarction | Atherosclerosis | Other acute ischaemic heart diseases | Heart failure | Stroke, not specified as haemorrhage or infarction | |
| **Kyrgyzstan** | Atherosclerotic heart disease | Stroke, not specified as haemorrhage or infarction | Other and unspecified cirrhosis of liver | Chronic ischaemic heart disease, unspecified | Atherosclerotic cardiovascular disease, so described | Other specified chronic obstructive pulmonary disease | Fetus and newborn affected by premature rupture of membranes | Acute myocardial infarction, unspecified | Cerebral atherosclerosis | Breast, unspecified | |
| **Egypt** | Heart failure | Essential (primary) hypertension | Cardiac arrest | Senility | Fibrosis and cirrhosis of liver | Hepatic failure, not elsewhere classified | Intracerebral haemorrhage | Other acute ischaemic heart diseases | Elevated blood glucose level | Respiratory failure, not elsewhere classified | Atherosclerosis |
| **Nicaragua** | Acute myocardial infarction, unspecified | Stroke, not specified as haemorrhage or infarction | Chronic renal failure, unspecified | Pneumonia, unspecified | Non-insulin-dependent diabetes mellitus with renal complications | Essential (primary) hypertension | Chronic obstructive pulmonary disease, unspecified | Cervix uteri, unspecified | Non-insulin-dependent diabetes mellitus without complications | Breast, unspecified | |
| **Tajikistan** | Essential (primary) hypertension | Senility | Stroke, not specified as haemorrhage or infarction | Other acute ischaemic heart diseases | Atherosclerosis | Unspecified diabetes mellitus | Acute myocardial infarction | Other ill-defined and unspecified causes of mortality | Chronic ischaemic heart disease | Complications and ill-defined descriptions of heart disease | |

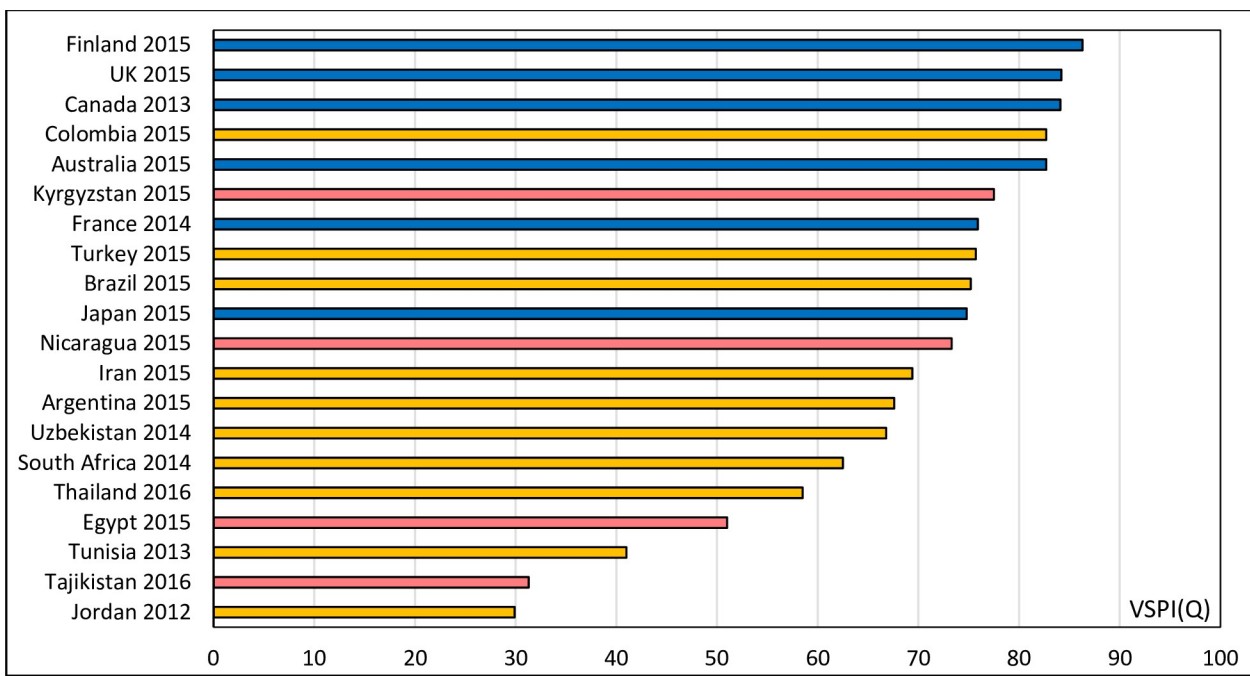

**Fig 4. Ranked scores of the Vital Statistics Performance Index for quality VSPI(Q), selected countries, c. 2015.** Low SDI, Middle SDI, High SDI.

dementia death rates at 85 years and above in reported COD data are, improbably, six times higher in Australia than in Japan, most likely resulting from different certification practices favoring the use of garbage coding in Japan [23].

Although High SDI countries on average had a lower proportion of garbage codes than Middle and Low SDI countries, the correlation was not very strong because of a number of outlier countries. For example, among the Middle and Low SDI countries Colombia, Kyrgyzstan and Nicaragua did better or as well as some of the High SDI countries. Furthermore, when we distinguished the garbage codes into high and low impact codes, surprisingly some low SDI countries (Kyrgyzstan and Nicaragua) had much lower (9–13%) high impact codes than several of the developed countries. The implication of this important finding is that targeted efforts to improve death registration completeness and minimize the use of garbage codes in COD data are possible to implement at comparatively low or medium levels of development, making the mortality data system much more useful for guiding public policy.

Both the pattern and volume of garbage codes among the 10 Middle SDI countries varied substantially. In Argentina, Tunisia, Thailand and South Africa, half of all deaths are being assigned to garbage codes, compared with less than one quarter in Colombia and Uzbekistan. A similar variation was found between the four Low SDI countries where Kyrgyzstan and Nicaragua had less than 25% of their deaths being assigned a garbage code compared to two thirds (65%) in Egypt and half (51%) for Tajikistan. Importantly, when garbage codes commonly appear among the top 10–20 leading causes of deaths, they diminish the policy value of the data by underestimating the true impact of other leading causes. Our findings reveal that in some countries up to 7 out of 10 leading causes are in fact high impact garbage codes, providing no useful information for guiding policy.

These findings are at the same time surprising and alarming. Countries spend considerable resources on maintaining their routine mortality surveillance systems. Improving completeness of death registration to ensure accurate all-cause mortality data is important to reliably

monitor trends in mortality by age and sex. At the same time, accurate cause-specific mortality data are fundamental for guiding the formulation and evaluation of interventions to reduce mortality and premature deaths. Only by investigating the specific diagnostic practices of individual countries, along with knowledge of the proportions of hospital and community deaths, would it be possible to comment on what leads to poor diagnostic practices in many countries. As our analysis demonstrates, many countries do not derive maximum policy benefit from these data due to the high, and in some cases, very high, prevalence of severe garbage codes. This, in part, could be due to lack of understanding and appreciation of the importance among certifying doctors of the public health value of correctly certified cause of death data, reflecting in turn the inadequate training many of them are receiving in how to certify correctly causes of deaths. To decrease the amount of garbage codes, effective strategies are required to train doctors in correct medical certification and in understanding why doing so is critically important for improving the population's health, as well as using automated verbal autopsy for those community deaths that cannot be medically certified. Another contributing factor to the higher proportions of garbage codes observed in some countries is likely to be the higher proportion of community deaths occurring without medical assistance, particularly in Low and Middle SDI countries. This need not automatically be the case, however. In Greenland, where about 10% of all deaths occur in remote small settlements with no physician present, these deaths are certified by a nurse, a health worker or another official and reported to the Chief Medical Officer who assigns the final ICD code. As a result, the VSPI for Greenland was higher than one might have expected, with a medium quality performance score of 66% [24]. However globally, the proportion of community deaths is estimated to be about 2/3rds of all deaths, most of which in low-income countries are not medically certified and therefore more likely to end up with a garbage code [25].

Although our results confirmed that there is an inverse relationship between the SDI level and amount of garbage codes, they also showed that some countries, despite relatively low socio-economic development, have managed to develop their vital statistics systems sufficiently to provide data that are fit for purpose. Five countries classified as being of middle or low SDI (Turkey, Brazil, Colombia, Kyrgyzstan and Nicaragua) had VSPI(Q) scores high enough to be considered to have well performing reporting systems. These five countries have invested in improving the quality of their mortality reporting systems [4]. Much could be learned from their experiences about what strategies were used to ensure that their systems provide policy relevant data to improve population health and survival. Conversely, Jordan, Tunisia and Tajikistan returned the lowest VSPI(Q) scores suggesting that their systems will need considerable improvement, both in COD quality and in completeness.

Interestingly, although there was some variation across countries in the number and ranking of garbage codes, they were remarkable similar. For instance, all were misdiagnosed non-communicable diseases, suggesting that the countries in our sample were all reasonably well advanced in their epidemiological transition. Heart failure, Senility and Other ill-defined causes were commonly used garbage codes in all countries, irrespective of their SDI level, with the only difference being where they appeared in the ranking among the leading causes of death. For example, Heart Failure was often ranked as the top leading cause in Low SDI countries, while in the Middle and High SDI countries it appeared at the 3rd or 4th rank. But the most notable difference was that Low and Middle SDI countries had many more garbage codes among the leading causes, and particularly those having the greatest impact for policy such as Senility, Hypertension and Other ill-defined.

A limitation of this study comes from the fact that it only investigates the output of the mortality system and not the amalgam of procedures and practices that collectively produce the data; hence conclusions about what underlies the observed differences are mostly speculative.

Another limitation was the small number of Low SDI countries included, which was due to lack of publicly available COD data. Further, data sets for most countries are at least 5 years old (2012–16) and may not adequately reflect improvements in the interim in both the mortality and the socio-economic situation of the country, which was assessed based on 2017 data.

An unexpected finding was that garbage codes were common, not only at the older ages, but worryingly constituted sizeable proportions in most age groups including children and adolescents under 20 years. The implication is that the evidence base for correctly understanding which are the leading causes of deaths in different age groups and for guiding health interventions designed to prevent premature deaths is likely to be significantly distorted by poor diagnostic practices. More reliable cause of death data will better inform debates about health sector priorities and strengthened health system responses, which can be expected to lead to better health and survival.

## Supporting information

**S1 File. What is Anaconda?.**
(DOCX)

**S2 File. ICD-10 composition of garbage codes.**
(DOCX)

**S3 File. Country characteristics.**
(DOCX)

**S4 File. Garbage codes by age groups.**
(DOCX)

## Author Contributions

**Conceptualization:** Kim Moesgaard Iburg, Lene Mikkelsen, Tim Adair, Alan D. Lopez.

**Formal analysis:** Kim Moesgaard Iburg, Lene Mikkelsen, Tim Adair.

**Investigation:** Kim Moesgaard Iburg.

**Methodology:** Kim Moesgaard Iburg, Lene Mikkelsen, Tim Adair, Alan D. Lopez.

**Project administration:** Kim Moesgaard Iburg.

**Software:** Kim Moesgaard Iburg, Lene Mikkelsen.

**Supervision:** Alan D. Lopez.

**Writing – original draft:** Kim Moesgaard Iburg, Lene Mikkelsen, Tim Adair.

**Writing – review & editing:** Kim Moesgaard Iburg, Lene Mikkelsen, Tim Adair, Alan D. Lopez.

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
