## [Decision Letter · Decision Letter 0]

21 May 2020

PONE-D-20-11375

Are cause of death data fit for purpose? Evidence from 20 countries at different levels of socio-economic development

PLOS ONE

Dear Dr. Iburg,

Thank you for submitting your manuscript to PLOS ONE. After careful consideration, we feel that it has merit but does not fully meet PLOS ONE’s publication criteria as it currently stands. Therefore, we invite you to submit a revised version of the manuscript that addresses the points raised during the review process.

A **rebuttal letter** that responds to **EACH** point raised by the academic editor and reviewer(s). You should upload this letter as a separate file labeled 'Response to Reviewers'.A **marked-up copy** of your manuscript that highlights changes made to the original version. You should upload this as a separate file labeled 'Revised Manuscript with Track Changes'.An **unmarked version** of your revised paper without tracked changes. You should upload this as a separate file labeled 'Manuscript'.

We look forward to receiving your revised manuscript.

Kind regards,

Brecht Devleesschauwer

Academic Editor

PLOS ONE

Additional Editor Comments:

In your revision note, please include EACH comment of the reviewers, provide your reply, and when relevant, include the modified/new text (or motivate why you decided not to modify the text). Note that failure to do so may still result in a rejection of the manuscript.

Journal Requirements:

2. Please include your tables as part of your main manuscript and remove the individual files.

Please note that supplementary tables should be uploaded as separate "supporting information" files.

Reviewers' comments:

Reviewer's Responses to Questions

**Comments to the Author**

1. Is the manuscript technically sound, and do the data support the conclusions?

Reviewer #1: Yes

Reviewer #2: Yes

2. Has the statistical analysis been performed appropriately and rigorously? 

Reviewer #1: N/A

Reviewer #2: Yes

3. Have the authors made all data underlying the findings in their manuscript fully available?

Reviewer #1: Yes

Reviewer #2: No

4. Is the manuscript presented in an intelligible fashion and written in standard English?

Reviewer #1: Yes

Reviewer #2: Yes

5. Review Comments to the Author

Reviewer #1: Many thanks for the opportunity to review your manuscript. I appreciate the effort that has gone into producing it, and its message is important. I believe this manuscript is publishable following some revisions and considerations that I would urge you to incorporate into a revised manuscript.

Abstract

1. Include in abstract methods that GCs were classified into four groups based on extent of coding error.

Introduction –

1. Include a sentence to reflect that GCs are also commonly referred to as ill-defined deaths.

Methods –

1. Lines 68-71 – Make clear what iteration of GBD was used, e.g. GBD 2017. Also cite GBD SDI reference ‘Global Burden of Disease Collaborative Network. Global Burden of Disease Study 2017 (GBD 2017) Socio-Demographic Index (SDI) 1950–2017. Seattle, United States: Institute for Health Metrics and Evaluation (IHME), 2018.’

2. Lines 72-80 – It would be helpful to include an example within each group to help the reader understand an applied case.

3. Line 84 – Explicitly mention GBD 2017.

4. Information is required on how the R-squared was derived that is discussed in the results section, and what is represents.

Results –

1. I noticed there was a discussion over the R-squared, what about gradient of linear regression to make a quantitative assessment of the relationship?

2. Line 106 – Embed into the paragraph below, perhaps after the first sentence? “Further insights into the general characteristics… can be found in…”.

3. Include a scatter plot, colour coding based on SDI.

4. Line 119 – Misspelling of ‘code’.

Discussion –

1. It would be helpful to discuss the differential risk of certain causes of death between SDI regions (for example NCDs vs Communicable disease) and whether they likely impacted the extent of GC error (e.g. severe consequences for PH decision making). Lines 168-172 does hint at this.

2. Deaths at home were mentioned in the abstract conclusion, it would be transparent to also include this in the discussion section.

Figures –

1. Include x- and y-axis labels.

2. Figure 3 – there is no need to have decimal places on this axis.

Supplementary file 3 –

1. Please consider removing the colour coding for the SDI groups and make the table similar to a heatmap to present a better visual of the data, whilst retaining the underlying data.

2. Include a header on the table illustrating that the columns are age-groups (in years).

3. The female table includes commas, rather than decimal points.

Reviewer #2: This paper provides a useful assessment of death registration systems in 20 countries using the same tool (ANACONDA), looking in particular at the quality of cause-of-death data. The paper presents some interesting results, in particular, when questioning the assertion that the proportions of garbage codes generally increase with age. The paper also demonstrates that there remains huge scope for improvement in death registration systems in almost all the countries considered. I have a few major comments that I hope will serve to strengthen the article.

Major comments:

- Could you clarify how the 20 countries were selected for this analysis? Did you simply pick some countries across the different levels of SDI?

- Anaconda seems to be widely used by countries and in regional workshops and has already lead to many publications, but it is still not in the public domain. Why not release an online version or a temporary version that users could download? Potential users should at least know how they can obtain the software and under what conditions.

- Although other articles specify how the garbage codes have been classified into four groups, readers should not have to read these previous publications to understand what these groups refer to. Could you elaborate a bit more on how GC are classified and provide some examples for example with a table? Can you also provide in the appendix the full list of garbage codes used and their classification?

- Likewise, the Adair-Lopez method is detailed in another PLOS ONE paper, but here the authors do not provide any explanation on what the method does and its limitations. They could perhaps mention that this method should ideally be triangulated with death distributions methods when applied at the national level. My recollection is that the method does not work well in settings where adult mortality is partly disconnected from child mortality, such as in South Africa and Colombia, two countries included here.

- What is the rationale for standardizing by age with the overall GBD age distribution of deaths? In assessing whether death registration systems are useful for health planning in their national context, would it be more relevant to look at the rank of causes of death without standardization?

- Finally, the article shows the heterogeneity of situations but provides few keys to understanding these variations. Could the fact that some countries have higher proportions of garbage codes be explained by a higher share of deaths occurring without medical assistance? The article would be even more useful if it further detailed the possible reasons for the variations in GC, perhaps using a few examples (France vs. Finland or Kyrgyzstan vs. Turkey).

Minor comments

-"Individual countries" in the abstract - do we need "individual"?

- In the abstract, "some low SDI countries, have vital statistics systems", please remove the comma.

- In the abstract, "half of all mortality data collected is of no use in guiding public policy", but this refers to cause-specific mortality. All-cause mortality remains hugely important to monitor trends in mortality by age. Perhaps this could be clarified in the discussion as improving the completeness of death registration must remain a priority even in the absence of an effective system for assigning causes of death.

- Line 119: garbage cods

- "deaths of little or no policy value" - strange expression, please consider revising.

6. PLOS authors have the option to publish the peer review history of their article (what does this mean?). If published, this will include your full peer review and any attached files.

Reviewer #1: No

Reviewer #2: Yes: Bruno Masquelier

---

## [Author Response · Author response to Decision Letter 0]

30 Jun 2020

Reviewer #1: 

Many thanks for the opportunity to review your manuscript. I appreciate the effort that has gone into producing it, and its message is important. I believe this manuscript is publishable following some revisions and considerations that I would urge you to incorporate into a revised manuscript.

Response: Thank you for this positive appraisal of the manuscript and its importance.

Abstract

1. Include in abstract methods that GCs were classified into four groups based on extent of coding error.

Response: Coders can only code what they see on the death certificate so it is actually certification errors. We have added a sentence in the Abstract to include this point.

Introduction –

1. Include a sentence to reflect that GCs are also commonly referred to as ill-defined deaths.

Response: In fact, GCs as defined in the paper, and applied in the ANACONDA tool, include more than ill-defined deaths since these deaths only comprise a part, albeit important, of the universe of causes that cannot, or should not be used to specify the underlying cause of death.

We have clarified this in the Introduction. 

Methods –

1. Lines 68-71 – Make clear what iteration of GBD was used, e.g. GBD 2017. Also cite GBD SDI reference ‘Global Burden of Disease Collaborative Network. Global Burden of Disease Study 2017 (GBD 2017) Socio-Demographic Index (SDI) 1950–2017. Seattle, United States: Institute for Health Metrics and Evaluation (IHME), 2018.’

Response: We used the GBD 2017 iteration, i.e. the latest available published in the Lancet in 2018, we have mentioned the GBD 2017 Study used in the Data and Methods section where we also have the SDI reference.

2. Lines 72-80 – It would be helpful to include an example within each group to help the reader understand an applied case.

Response: We have added these examples to the text, as requested.

3. Line 84 – Explicitly mention GBD 2017.

Response: This has been done.

4. Information is required on how the R-squared was derived that is discussed in the results section, and what is represents.

Response: The coefficient of determination R2 was calculated to measure the percentage of variation in garbage codes across all countries together. This relationship is mentioned together with the results of R2 derived from the regression line in the new Figure 2.

Results –

1. I noticed there was a discussion over the R-squared, what about gradient of linear regression to make a quantitative assessment of the relationship?

Response: We have added a brief phrase on the interpretation and mentioned the importance of the gradient for the relationship we identified.

2. Line 106 – Embed into the paragraph below, perhaps after the first sentence? “Further insights into the general characteristics… can be found in…”.

Response: This has now been done (now line 147-48).

3. Include a scatter plot, colour coding based on SDI.

Response: That is a helpful suggestion, thank you. We have now added a new Figure 1 scatterplot showing the percentage of total garbage codes versus country by their SDI value. Also added to the scatterplot is the simple linear regression line and R-squared value.

4. Line 119 – Misspelling of ‘code’.

Response: Now corrected.

Discussion –

1. It would be helpful to discuss the differential risk of certain causes of death between SDI regions (for example NCDs vs Communicable disease) and whether they likely impacted the extent of GC error (e.g. severe consequences for PH decision making). Lines 168-172 does hint at this.

Response: We have clarified this differential cause of death risk in the text, and expanded on the comparative importance of GC errors on the cause of death pattern at different levels of SDI.

2. Deaths at home were mentioned in the abstract conclusion, it would be transparent to also include this in the discussion section.

Response: We have expanded on this concept in the Discussion section and added a recent example from the literature.

Figures –

1. Include x- and y-axis labels.

Response: This is now done.

2. Figure 3 – there is no need to have decimal places on this axis.

Response: Corrected.

Supplementary file 3 –

1. Please consider removing the colour coding for the SDI groups and make the table similar to a heatmap to present a better visual of the data, whilst retaining the underlying data.

Response: Our understanding is that the table is already in the form of a heatmap. We also believe that the colour coding for SDI groups helps the reader to better understand the differential patterns experienced by different SDI groups.

2. Include a header on the table illustrating that the columns are age-groups (in years).

Response: Done.

3. The female table includes commas, rather than decimal points.

Response: Corrected.

Reviewer #2: 

This paper provides a useful assessment of death registration systems in 20 countries using the same tool (ANACONDA), looking in particular at the quality of cause-of-death data. The paper presents some interesting results, in particular, when questioning the assertion that the proportions of garbage codes generally increase with age. The paper also demonstrates that there remains huge scope for improvement in death registration systems in almost all the countries considered. I have a few major comments that I hope will serve to strengthen the article.

Response: Thank you for the generally positive assessment of the importance of our study. 

Major comments:

- Could you clarify how the 20 countries were selected for this analysis? Did you simply pick some countries across the different levels of SDI?

Response: We have clarified better the choice of countries in the text. As you know, all WHO member countries with cause of death data are asked to provide mortality information to WHO coded to at least the 3rd digit of ICD. Not many low-income countries, however, collect mortality data and hence our choice was limited to those who do and supply these data to WHO. 

- Anaconda seems to be widely used by countries and in regional workshops and has already lead to many publications, but it is still not in the public domain. Why not release an online version or a temporary version that users could download? Potential users should at least know how they can obtain the software and under what conditions.

Response: We have had the same request from many potential users. As a result, ANACONDA was made publicly available on June 10th. It can now be downloaded from the following link: www.crvsgateway.info/ANACONDA

- Although other articles specify how the garbage codes have been classified into four groups, readers should not have to read these previous publications to understand what these groups refer to. Could you elaborate a bit more on how GC are classified and provide some examples for example with a table? Can you also provide in the appendix the full list of garbage codes used and their classification?

Response: As requested, we have now clarified the basis and characteristics of the four-part categorization of garbage codes used in ANACONDA in the text.

We have now added a new Supporting information file 2 with the complete list of garbage codes identified by ANACONDA with their classification on 4 levels. 

- Likewise, the Adair-Lopez method is detailed in another PLOS ONE paper, but here the authors do not provide any explanation on what the method does and its limitations. They could perhaps mention that this method should ideally be triangulated with death distributions methods when applied at the national level. My recollection is that the method does not work well in settings where adult mortality is partly disconnected from child mortality, such as in South Africa and Colombia, two countries included here.

Response: This is a good suggestion. We have added a brief para in the methods section to explain how best to interpret the results of the Adair-Lopez method as used in ANACONDA, and also to clarify some of its limitations, particularly for populations where adult mortality is disconnected from child mortality, such as countries affected by high HIV prevalence.

- What is the rationale for standardizing by age with the overall GBD age distribution of deaths? In assessing whether death registration systems are useful for health planning in their national context, would it be more relevant to look at the rank of causes of death without standardization?

Response: We agree with the reviewer and in fact, we report on both standardized and unstandardized results. The reviewer is correct in that the latter are certainly more relevant for guiding national health policy and the need for health services. The point of age-standardising the universe of garbage codes was to investigate whether countries with a relatively old age structure, and hence relatively high average age at death, might have a greater fraction of garbage codes simply because of the higher likelihood of multiple co-morbidity in the elderly makes it more difficult to certify which was the underlying cause of death.

Our findings using age-standardisation suggest that this is not the case and only small differences were seen. 

We have clarified the purpose of the age-standardisation in the text.

- Finally, the article shows the heterogeneity of situations but provides few keys to understanding these variations. Could the fact that some countries have higher proportions of garbage codes be explained by a higher share of deaths occurring without medical assistance? The article would be even more useful if it further detailed the possible reasons for the variations in GC, perhaps using a few examples (France vs. Finland or Kyrgyzstan vs. Turkey).

Response: This is a good point. Without knowing the specific diagnostic practices of individual countries, it is hard to be very specific about the reasons for so much garbage coding in the data, although a common factor is surely the lack of appreciation, and quite possibly training, among physicians about how to correctly certify causes of death and why doing so is critically important for improving the population’s health. We have added some additional text and a recent example from our research team analysing quality of COD data in Greenland to the Discussion to better emphasise this point. The reviewer is correct in suggesting that the higher the proportion of community deaths that occur without medical assistance, the higher will be the fraction of garbage codes in the data. Nonetheless, some countries have accorded high priority to improving the registration and certification of deaths, given the critical policy value of these data. We have already mentioned this in the Discussion, with Turkey, Brazil, Colombia, Kyrgyzstan and Nicaragua being five countries that have invested in improving the quality of their mortality reporting systems, but added the above issues now to the discussion. 

Minor comments

-"Individual countries" in the abstract - do we need "individual"?

Response: Now corrected.

- In the abstract, "some low SDI countries, have vital statistics systems", please remove the comma.

Response: Done

- In the abstract, "half of all mortality data collected is of no use in guiding public policy", but this refers to cause-specific mortality. All-cause mortality remains hugely important to monitor trends in mortality by age. Perhaps this could be clarified in the discussion as improving the completeness of death registration must remain a priority even in the absence of an effective system for assigning causes of death.

Response: This is a good point. We have amended the Abstract accordingly and made mention of the policy value of good all-cause mortality data in the Discussion, as suggested. We have also added a justification or implication of our research to the Introduction and Discussion.

- Line 119: garbage cods

Response: This has now been corrected.

- "deaths of little or no policy value" - strange expression, please consider revising.

Response: This has now been corrected.

---

## [Decision Letter · Decision Letter 1]

29 Jul 2020

Are cause of death data fit for purpose? Evidence from 20 countries at different levels of socio-economic development

PONE-D-20-11375R1

Dear Dr. Iburg,

We’re pleased to inform you that your manuscript has been judged scientifically suitable for publication and will be formally accepted for publication once it meets all outstanding technical requirements.

Kind regards,

Brecht Devleesschauwer

Academic Editor

PLOS ONE

Additional Editor Comments (optional):

Both reviewers were satisfied with the revisions made by the authors. Reviewer #2 identified some typos which can be corrected in the final mansuscript.

Reviewers' comments:

Reviewer's Responses to Questions

**Comments to the Author**

1. If the authors have adequately addressed your comments raised in a previous round of review and you feel that this manuscript is now acceptable for publication, you may indicate that here to bypass the “Comments to the Author” section, enter your conflict of interest statement in the “Confidential to Editor” section, and submit your "Accept" recommendation.

Reviewer #1: All comments have been addressed

Reviewer #2: All comments have been addressed

2. Is the manuscript technically sound, and do the data support the conclusions?

Reviewer #1: Yes

Reviewer #2: Yes

3. Has the statistical analysis been performed appropriately and rigorously? 

Reviewer #1: N/A

Reviewer #2: Yes

4. Have the authors made all data underlying the findings in their manuscript fully available?

Reviewer #1: (No Response)

Reviewer #2: Yes

5. Is the manuscript presented in an intelligible fashion and written in standard English?

Reviewer #1: Yes

Reviewer #2: Yes

6. Review Comments to the Author

Reviewer #1: (No Response)

Reviewer #2: Many thanks for the responses to the comments and suggestions from the reviewers. All comments have been addressed.

two typos:

- line 94 : three broad cause groups

- line 388: evidence base is

7. PLOS authors have the option to publish the peer review history of their article (what does this mean?). If published, this will include your full peer review and any attached files.

Reviewer #1: No

Reviewer #2: **Yes: **Bruno Masquelier

---

## [Editor Report · Acceptance letter]

3 Aug 2020

PONE-D-20-11375R1 

Are cause of death data fit for purpose? Evidence from 20 countries at different levels of socio-economic development 

Dear Dr. Iburg:

I'm pleased to inform you that your manuscript has been deemed suitable for publication in PLOS ONE. Congratulations! Your manuscript is now with our production department. 

Kind regards, 

on behalf of

Prof. Dr. Brecht Devleesschauwer 

Academic Editor

PLOS ONE